# Advances in Natural or Synthetic Nanoparticles for Metastatic Melanoma Therapy and Diagnosis

**DOI:** 10.3390/cancers12102893

**Published:** 2020-10-09

**Authors:** Maria Beatrice Arasi, Francesca Pedini, Sonia Valentini, Nadia Felli, Federica Felicetti

**Affiliations:** Department of Oncology and Molecular Medicine, Istituto Superiore di Sanità, 00161 Rome, Italy; mariabeatrice.arasi@guest.iss.it (M.B.A.); francesca.pedini@iss.it (F.P.); sonia.valentini@guest.iss.it (S.V.); federica.felicetti@iss.it (F.F.)

**Keywords:** nanoparticles, exosomes, extracellular vesicles, melanoma, therapy, diagnosis, prognosis

## Abstract

**Simple Summary:**

Malignant melanoma is the most aggressive skin cancer; its incidence is constantly growing in the white population. In the advanced stage of the disease, after metastatic dissemination, patients have a poor prognosis. Nanomedicine represents a new frontier in cancer treatment; in this field, synthetic and natural nanoparticles (NPs) may represent an important therapeutic and diagnostic opportunity. This review provides an overview of current knowledge in this area: several kinds of NPs, like PLGA, chitosan, liposome and gold-NPs, are used to increase the specificity of drug delivery, allowing a dose reduction and, consequently, a lower toxic effect. Particular attention is given to exosomes (EXOs), an example of natural NPs, important both in conveying molecules with a therapeutic function and in the diagnostic field.

**Abstract:**

Advanced melanoma is still a major challenge in oncology. In the early stages, melanoma can be treated successfully with surgery and the survival rate is high, nevertheless the survival rate drops drastically after metastasis dissemination. The identification of parameters predictive of the prognosis to support clinical decisions and of new efficacious therapies are important to ensure patients the best possible prognosis. Recent progress in nanotechnology allowed the development of nanoparticles able to protect drugs from degradation and to deliver the drug to the tumor. Modification of the nanoparticle surface by specific molecules improves retention and accumulation in the target tissue. In this review, we describe the potential role of nanoparticles in advanced melanoma treatment and discuss the current efforts of designing polymeric nanoparticles for controlled drug release at the site upon injection. In addition, we highlight the advances as well as the challenges of exosome-based nanocarriers as drug vehicles. We place special focus on the advantages of these natural nanocarriers in delivering various cargoes in advanced melanoma treatment. We also describe the current advances in knowledge of melanoma-related exosomes, including their biogenesis, molecular contents and biological functions, focusing our attention on their utilization for early diagnosis and prognosis in melanoma disease.

## 1. Introduction

Advanced cutaneous melanoma is a highly aggressive and drug-resistant cancer [1,2]. The main problem associated with treating melanoma is its low response rate to conventional therapy. Nevertheless, the rapidly developed fields of melanoma research are introducing new potential therapeutic approaches, focusing on developing new efficient drugs and delivery systems. Before 2011, metastatic melanoma was considered incurable, leading to death within 18 months of diagnosis [3,4]. The therapies used were based on the administration of chemotherapeutic agents such as dacarbazine and temozolomide [5]. In order to circumvent the intrinsic resistance against chemotherapy, researchers explored the possibility of potentiating the immune response by using immune stimulators, such as interleukin 2 (IL-2), interferon alpha (IFN-α), ipilimumab and thymosin alpha 1 [6,7]. Afterwards, as result of genome sequencing and identification of the main mutation, a series of therapeutic agents have been discovered; about half of all melanomas have mutations in the BRAF (v-raf murine sarcoma viral oncogene homolog B1) gene [8,9]. These mutations constitutively activate the MAPK (mitogen-activated protein kinase) pathway, resulting in uncontrolled cell proliferation, tumor development and growth [1,10]. Several tyrosine kinase inhibitors (TKIs) have been developed that block the MAPK pathway and inhibit cell growth and survival. The use of TKIs has shown a significant improvement of patient overall survival and the combinations therapy with BRAF/MEK inhibitors is now accepted as the standard care for BRAF-muted advanced melanoma. Unfortunately, in most treated patients, responses to these inhibitory molecules are transient, and, in a few months, a therapy-resistant phenotype occurs. Recently, the inhibition of the programmed cell death 1 (PD-1)/programmed cell death 1 ligand 1 (PD-L1) axis, a suppressor of T-cell response, turned out to be an effective therapeutic strategy with a low toxicity profile and contributed to better clinical outcomes compared to chemotherapy and also to other immune mediated approaches with cytotoxic T lymphocyte-associated 4 (CTLA-4) inhibition [11,12,13]. However, substantial heterogeneity exists in metastatic melanoma response to treatments; therefore, the development of multimodality strategies is the major challenge to fight this deadly disease. Several types of nanoparticles (NPs) and nanovesicles have been explored for their employment as drug carriers in melanoma therapy and their functionalization with specific molecules, such as antibodies, can generate different smart nanodrugs for application in melanoma therapy (Figure 1). Herein, we will discuss the new frontiers in the treatment of patients with unresectable or metastatic melanoma, paying special attention to new anticancer molecules and their delivery through natural or synthetic nanoparticles.

## 2. Nanoparticles and Drug Delivery

Nanomaterials are delivery vehicles used to protect the intended drug against degradation and enhance its stability. Using a nanotechnology treatment for several types of cancer enhances targeted delivery to cancer cells and uptake, as well as reducing cytotoxic side effects on normal tissues [14].

Chemotherapeutic agents generally exhibit a short plasma half-life due to rapid clearance and fast biodegradation; these elements, together with the low specificity, make it necessary to use high doses, resulting in severe toxic effects. In addition, since cancer diagnosis is often late, cancer cells have already invaded other parts of the body [15].

Nanotechnologies are based on the use of different polymeric systems whose modifiable physical and chemical systems provide tools to improve different aspects of the therapy. Some examples are the release of the drug in a tissue-specific way, the decrease in side effects, the protection from drug degradation, the possibility to reduce the frequency of administrations and the toxicity of the substance itself [16]. Depending on their specific structural characteristics, nanoparticles can easily penetrate tissues and cells, circumscribe the biological effect of the drug on a specific cell type and modify the pharmacokinetic properties, thus allowing a prolonged drug release over time. The Food and Drug Administration (FDA) approved only a few nanodrugs for use and clinical trials, as they have been shown to either target and directly kill tumor cells or improve, overall, targeted chemotherapy drug delivery. The formulation of these nanoparticles increases the drug concentration into the tumor by 100–400% [17].

### 2.1. PLGA

The FDA and the European Medicine Agency (EMA) approved PLGA nanoparticles (PLGA-NPs) in 2011 as a system for conveying nucleic acids, drugs and other molecules for cancer and other diseases. PLGA is a biocompatible and biodegradable polymer (poly-lactic acid and poly-glycolic acid) whose degradation produces non-toxic substances, such as water and carbon dioxide [18]. The PLGA-NPs’ cellular uptake depends on both the size of the NP and the composition of the plasma membrane of the cells themselves [19]. The different physical characteristics of PLGA-NPs, such as their size and morphology, can be determined by varying the parameters of the synthesis method used. Nanoparticles have a large surface area that can be functionalized with different agents to make the vehicle target-specific [20].

In the last period, several publications highlighted the efficiency of PLGA-NPs in delivering anticancer agents; in a recent work, Arruda and his co-workers showed that the encapsulation of a peptide with antitumoral activity, P20 (CSSRTMHH), within the PLGA-NPs conjugated to the C peptide with a homing function (CVNHPAFACGYGHTMYYHHYQHHL), strongly increases their functionality. The therapeutic effect of these NPs was evaluated in a syngeneic model of metastasis and was shown to produce the same inhibitory effect of free P20 at fivefold higher concentration, reducing by 28% the number of lung nodules [21].

In another recent study, trastuzumab has been used as a recognizing unit attached to NPs for Her-2 positive breast cancer therapy. Targeted nanoparticles were prepared by combining PLGA/ polyethylenimine (PEI)/lipid nanoparticles (PPLNs) with trastuzumab through the electrostatic adsorption method to deliver the antitumor drug docetaxel. The results showed that trastuzumab-PPLNs had a cell-targeting effect and could effectively inhibit the proliferation of cancer cells. This work demonstrates that trastuzumab-PPLNs are a promising treatment for breast cancer and can be considered a proof of concept for anticancer therapies, including melanoma [22].

Following a similar approach, Kou et al. selected paclitaxel as anti-metastatic and anti-angiogenic model drug to produce paclitaxel-loaded PLGA-NPs (Ptx-NPs), coated with a polylysine cationic SM5-1 single-chain antibody (Ptx-NP-S) which binds to a membrane protein specifically expressed in melanoma, hepatocellular carcinoma (HCC) and breast cancer cells. The authors demonstrated that Ptx-NP-S not only showed very high affinity to SM5-1 but also was internalized by the target cells. They proved a significantly enhanced cytotoxicity of Ptx-NP-S when compared to non-targeted paclitaxel-loaded PLGA-NPs, suggesting SM5-1 single-chain antibody-polylys as a good candidate to synthesize cancer-targeted PLGA nanoparticles [23].

Another important field of application of PLGA-NPs is as peptide transporters in vaccines. Vaccines could be promising tools for treating cancers, including melanoma [24]; however, several studies obtained unsatisfying results, possibly because of the low stability of peptides and delivery approaches. Thus, the appropriate and efficient peptide carrier systems still continue to be one of the major obstacles. In order to overcome this difficulty, Li and coworkers recently used PLGA-NPs as a new method to deliver tumor-associated antigen (TAA) peptides to dendritic cells (DC) [25].

### 2.2. Chitosan

Chitosan is a bio-poly-amino-saccharide cationic polymer obtained from chitin deacetylation moieties formed by the repetition of two units: N-acetyl-D-glucosamine and D- glucosamine [26]. The molecular weight and the extent of deacetylation of the chitosan influence its properties; this is employed specially to prepare micro and nanoparticles capable of conveying nucleic acids or medicines. The chitosan backbone consists of multiple free amino and hydroxyl groups that can be functionalized. Versatile approaches have been reported for the construction of chitosan-based nanomaterials, such as nanogels, NPs, micelles, liposomes, nanofibers and nanospheres [27]. The polymer’s success is due to its properties such as: biocompatibility and biodegradability, FDA approval in clinical use, the possibility of modifying its surface and to complex it in nanoparticles conveyed against cells and/or target tissues. Chitosan exhibits no or minimal toxicity, which made it widely regarded as a safe material in drug delivery [28]. Various cargos, hydrophilic and also hydrophobic, can be incorporated into chitosan-NPs. Their release is influenced by solubility, diffusion, and size of the nanoparticles [29]. Many publications show innovative ways for antitumor drug delivery based on environmental response and targeting principles [30,31].

A recent study evaluated the effect of doxorubicin encapsulated into chitosan/alginate nanoparticles to inhibit viability and tumor growth of melanoma cells in a mouse melanoma model. The experiments revealed that the encapsulation of doxorubicin could be considered advantageous because it increases intracellular accumulation and produces a long-lasting cytotoxic effect [32]. The cationic nature of chitosan makes it suitable to complex anionic nucleic acids, protecting them from degradation by nucleases in serum [33].

One of the possible applications of this property is the transport of two specific categories of non-coding RNAs called microRNA (miRs) and short interfering RNA (siRNAs) [34,35]. MiRs regulate the expression of messenger RNAs (mRNAs) at a post-transcriptional level. They are implicated in various aspects of tumor progression, including proliferation, apoptosis, and epithelial–mesenchymal transition (EMT), and they can promote or suppress tumorigenesis and metastatization, thus acting either as onco-suppressors or oncogenes [14,36,37,38].

The possibility of protecting these molecules from enzymatic degradation using chitosan-NPs represents a very important therapeutic option. SiRNAs have broad potential as therapeutic agents because they are able to silence any gene in a reversible way. To achieve the clinical potential of siRNAs, delivery materials are required to transport them in the cells of target tissues.

Rostami and coworkers, in a recent study, developed the alginate-conjugated trimethyl chitosan (ATMC) nanoparticles loaded with specific siRNAs. In particular, in the cancer cells the activation of S1P/sphingosine-1-phosphate receptor 1 (S1PR1) and IL-6/glycoprotein 130 (GP130) genes induces STAT3 phosphorylation, which in turn stimulates their expression.

On this basis, the authors investigated the potential anticancer effect by silencing the S1PR1 and GP130 genes, through ATMC-NPs containing specific siRNAs; the results obtained in different cancer cell lines, including melanoma B16F10, although in need of further study, have been encouraging [35].

### 2.3. Liposomes

Liposomes are small lipid vesicles (50 to 100 nm) generated from cholesterol, which reduces their permeability and increases their in vivo and in vitro stability [39]. Liposomes, categorized by the composition and mechanism of intracellular delivery, fall under five types: conventional liposomes (CL), pH-sensitive liposomes, cationic liposomes, immunoliposomes, and long-circulating liposomes (LCL). Liposomes with modified surfaces have also been developed using several molecules, such as glycolipids or sialic acid [40]. To prolong their circulation time, liposome surfaces have been coated with a hydrophilic polymer, such as polyethylene glycol (PEG), thus increasing repulsive forces between liposomes and plasma components [41]. Liposomes seem to be an almost ideal drug-carrier system due to their morphology being similar to that of cellular membranes and for their ability to incorporate various substances.

Recently, Gao et al. demonstrated that liposome-PEI encapsulation improved the uptake of n-Butylidenephthalide (BP) through enhancement of cell endocytosis, indicating that the BP/lipo PEG PEI complex (LPPC) has great potential in melanoma treatment. In particular, because BP is metabolized very rapidly, its antitumor role, already demonstrated in various types of cancer, including melanoma, would be limited. To avoid this problem, the researchers developed the complex BP/LPPC, improving both the uptake of this drug by the cells and its stability over time [42].

Yin et al. prepared R8-dGR peptide modified paclitaxel (PTX) and hydroxychloroquine (HCQ) co-loaded liposomes (PTX/HCQ-R8-dGR-Lip) to enhance delivery through recognition of integrin αvβ3 receptors and neuropilin-1 receptors on B16F10 melanoma cells. The results showed that R8-dGR modified liposomes (R8-dGR-Lip) enhanced tumor-targeting delivery in vitro and in vivo [43].

Health authorities have approved different liposomal formulations of small molecule therapeutics; their ability to deliver macromolecules, such as nucleic acid-based therapeutics plasmid DNA (pDNA) and siRNA, to disease sites is under investigation. In a recent study, lipid nanocapsules were used to combine two anti-melanoma strategies, gene therapy by using Bcl-2 siRNA and ferrocifens as novel anticancer agents. The results showed a 50% reduction in tumor volume in tumor-bearing animals compared to the control group [44].

Sadhu et al. utilized liposomes to deliver glutathione disulfide to melanoma cells. These liposomes slowed down tumor proliferation by 85–90% in melanoma-bearing mice and significantly improved the survival rate [45]. Interestingly, lipid constituents of liposomal vesicles can be tailored to achieve particular immunogenic responses. Manipulation of surface charge density in cationic liposomes can regulate immune response [46].

Fu and coworkers developed paclitaxel (PTX)-loaded liposomes functionalized with trans activator of transcription (TAT), the most frequently used cell-penetrating peptide, and cleavable PEG. Under physiological conditions, the TAT peptide is shielded by a PEG layer and so liposomes exhibit a good stability in the bloodstream. When liposomes arrive at the tumor site, exogenous reducing agents could remove PEG, causing TAT exposure and facilitating cell internalization. In this study, the authors showed a tumor inhibition of around 70% in B16F1-bearing mice [47].

Lastly, the liposomes may also be used as an adjuvant to boost the protective or therapeutic immune response [48].

### 2.4. Gold Nanoparticles

Gold nanoparticles (AuNPs) are a type of nanoparticles produced in various sizes and shapes (i.e., gold nanospheres, nanorods, nanocages, nanoshells, and nanostars). They are dependent on the synthetic methods adopted for their preparation [49]. These nanoparticles can be prepared in a broad range of core sizes (1 to 150 nm), which makes it easier to control their dispersion. The presence of a negative charge on the surface of gold nanoparticles makes them easily modifiable.

Literature reports that various types of gold nanoparticles are effective tools in interfering with human cancer cell biology, including melanoma [50]. Indeed, their application in therapeutic development and in cancer diagnosis could provide a potential step forward in cancer theranostics [51]. Thanks to their physical, chemical and biological features, AuNPs can be used in imaging technology, such as computed tomography, ultrasound and magnetic resonance and a few of them have been approved by the FDA in medicine [52].

Kim et al. demonstrated the use of AuNPs as a contrast agent for the quantitative molecular combining of high-resolution photoacoustic tomography (PAT) of melanomas in vivo. To avoid the limitations of conventional diagnostic techniques, the authors exploited the enormous optical absorbing gold nanocages (AuNCs), obtaining a 300% increase in the contrast when AuNCs were bioconjugated with [Nle4,d-Phe7]-α-melanocyte-stimulating hormone compared to the control nanocages [53].

Through the addition of various biomolecules such as drugs, target ligands and genes, the nanoparticles could function in an enhanced way. To overcome in vivo delivery barriers, Au nanocomplexes are normally modified with functional moieties such as stabilizing materials, target ligands, and bio responsive linkers [54,55].

Conjugation with the anti-nucleosome monoclonal antibody (mAb 2C5) of PEGylated liposomes improves cancer cell-specific delivery targeting the matrix metalloprotease 2 in the tumor microenvironment [56]. Many studies showed that the use of magnetic fields or altered temperatures and light improve the delivery of gold particles including anticancer agents [57].

### 2.5. Other Nano-Agents

Other nanoparticles have a potential use as anti-melanoma agents (Table 1) [58,59,60,61,62,63,64,65].

### 2.6. Specific Drug Delivery

The ideal drug carrier would have the ability to deliver the drug to the cell/tissue/organ in a specific way to make the cancer treatment more efficient. A potentially powerful technology is represented by targeted nanoparticles. The most used approach is to conjugate to nanoparticles antibodies or antibody-derivatives which must be able to overcome the physiological barriers and conditions that limit access to the target, and to bind target cells through specific antibody–receptor interactions [66].

Primary tumors as well as metastatic tumors overexpress some antigens on their surfaces. Particularly, for melanoma treatment, the literature shows the specificity of many antigens.

Secreted protein acidic and rich in cysteine (SPARC) was deepened in the tumorigenicity and progression of melanoma, resulting as a nonstructural matricellular protein [67]. The conjugation of nanoparticles with a peptide able to specifically recognizing the SPARC protein, has made it possible to obtain an important tool that exhibited high tumor affinity with a very low binding in negative cell lines. In in vivo targeting experiments, an increase of SPARC-targeted nanoparticles was observed in tumor tissue as compared with the control group. Moreover, a very important result was the detection of both primary tumors and metastatic growth obtained by using the fluorescent version of these nanoparticles [68].

Melanoma antigen genes (MAGE)-A proteins are a melanoma-specific-antigen family [69]. Saeed and coworkers in a recent work described that the single-chain variable fragment (scFv) antibodies directed against MAGE-A1 peptide, presented by human leukocyte antigen A1 (HLA-A1), in short M1/A1, have been coupled to liposomes to achieve specific melanoma targeting. Through experiments by flow cytometry and confocal microscopy, the authors demonstrated the bind between M1/A1-positive and B-cells as well as with melanoma cells and the internalization by these cells [70].

The chondroitin sulfate proteoglycan 4 (CSPG4) is a highly specific marker of the nevomelanocyte lineage and has been utilized to target melanoma. In a recent work, Falvo and coworkers produced nanoparticles conjugated to a monoclonal antibody specific for CSPG4 encapsulating cisplatin molecules. Flow cytometry showed a specific binding to CSPG4 (+) melanoma cell lines. Selectivity towards melanoma was also confirmed in xenograft models: targeted nanoparticles significantly inhibited growth of melanoma xenografts whereas modestly affected breast carcinoma xenografts [71].

These studies highlight the importance of the potential active targeting to improve the outcome of a nanoparticle-based therapy.

### 2.7. Natural Nanoparticles

All cell types of the body release extracellular Vesicles (EVs), which are present in all body fluids. They are defined by their mechanism of cellular production and include exosomes (EXOs), microvesicles (MVs) and apoptotic bodies [72]. The diameter of the EXOs ranges from 30 to 150 nm; they are secreted by most cell types into the extracellular microenvironment under normal and pathological conditions and can be found in blood, sweat, saliva, urine, amniotic fluids, breast milk, cerebrospinal fluids and malignant ascites [73,74,75,76]. EXOs have a membrane consisting of a lipid bilayer and contain various biomolecules, including proteins, lipids, RNA and DNA, suggesting their involvement in the regulation of various biological functions through different molecular mechanisms [77,78,79].

When EXOs are taken up by other cells, their cargo (mRNAs, miRs and protein) is transferred and influences the phenotype of recipient cells [80,81]. EXOs mediate cell-to-cell communication and are involved in different physiological and pathological processes, also playing an important role in cancer [82]. Like other tumors, melanoma cells modify their microenvironment to enable growth and metastasis via direct cellular interactions or the indirect release of soluble factors and/or EXOs.

Similarly to synthetic nanoparticles, EXOs can be considered drug delivery systems and possess important advantages such as long-term circulatory capability, good biocompatibility, minimal toxicity and intrinsic ability to target specific cells [83]. Moreover, one of the most useful properties of EXOs is their ability to cross the cytoplasmic membrane [84] and the blood/brain barrier [85].

Altogether, these characteristics make EXOs a very interesting drug delivery system [86]. Recently, several studies have focused on specific modification of EXOs surface to make them a proper candidate for cellular uptake and target specificity [87]. In particular, it has been proved that modification of EXO–tetraspanin complexes strongly influences target cell selection both in vitro and in vivo [88], improving the targeting of tissues and cell types of interest. Other modified transmembrane proteins are the platelet-derived growth factor receptors [89]. In 2018, Kim et al. engineered macrophage-derived EXOs loaded with the anticancer drug paclitaxel to improve their circulation time in the blood and allow them to target pulmonary metastases. By using this modified EXOs, the drug can selectively deliver to target cancer cells and also can increase the survival rate in an in vivo lung cancer model [90].

In the last decade, several studies have focused on immune system stimulation for the treatment of different cancers. Dendritic cell-derived EXOs (dexosome) have been successfully engineered to target helper T cells to stimulate cytotoxic T cell proliferation, influence T cell differentiation and create an anti-tumor environment [91]. Dexosomes have entered clinical trials for colorectal cancer [92], non-small cell lung cancer [93] and metastatic melanoma [94]. Particularly in melanoma cancer, EXOs are studied as therapeutic cell-free vaccines, e.g., dendritic cell-derived EXOs loaded with MAGE3 antibodies were administered subcutaneously or intradermally to stimulate the immune response of melanoma stage IIIb/IV patients obtaining positive clinical effects [95].

Moreover, Zhu et al. analyzed the fundamental role of EXOs derived from natural killer cells to exert cytotoxic effects on melanoma cells, suggesting this could be a potential immunotherapeutic strategy for cancer [96]. More recently, Luigini L. et al. performed a homemade test for the analysis of natural killer exosomes derived from plasma of melanoma patients and healthy donors. Their results showed that these vesicles could be used to improve cancer treatment in combination with NK cells or immune checkpoint-based therapies [97].

Among the new therapeutic approaches, very interesting results were reviewed by Chillà A. et al. In their paper, they refer to the therapeutic use of some particular cells used as natural nanoparticles [98]. For example, the possibility to target an experimental melanoma with red blood cells (RBCs) and platelet membrane-coated gold nanoparticles containing curcumin was described. In this study, the natural cell membranes have been used as biointerfaces that interact with the host environment. Platelet membrane coating has been used to target cancer cells while RBC membrane coating has been used to provide clearance by macrophages [99]. This cell-mediated delivery would provide controlled dosing, no systemic toxicity, and high tolerance by the patient and could be possibly evaluated as a therapeutic strategy in melanoma [98].

### 2.8. Natural Nanoparticles for Metastatic Melanoma Diagnosis

Early diagnosis and effective therapy are undoubtedly the primary issues for prolonging the survival of cancer patients. At present, the solid biopsy used for diagnosis is mostly the basis for treatment of cancer. Unfortunately, this method has considerable limitations as it is invasive and sometimes not feasible. Accumulating evidence over the past few decades has highlighted the potential use of circulating molecular biomarkers as a promising platform for the noninvasive diagnosis and definition of prognosis. As previously reported, the cargos incorporated in EXOs reflect the contents and the pathological status or signaling alterations of the original cells. Moreover, the cargos are protected from degradation and are isolable from patients’ body fluids. Thus, analyses of circulating EXOs and their contents in blood could represent a promising strategy in cancer diagnosis advancement and monitor patients’ therapeutic response.

For instance, PD-L1 expression in tumor tissues is widely used in selecting patients who will benefit from treatment with immunotherapy. Recently, evaluating the role of exosomal PD-L1 expressed in melanoma patients treated with anti-PD-1/PD-L1 antibodies, Danesi R. and his group showed that exosomal PD-L1 expression increased in subjects with disease progression and decreased in patients responding to treatment, while no significant changes were observed in patients with disease stabilization [100]. In the same year, Chen G. et al. [101] showed that Interferon-γ (IFN-γ) up-regulates PD-L1-containing EXOs secretion and changes during the course of anti-PD-1 therapy. In accordance, a very recent prospective study to evaluate exosomal-PD-L1 expression from 100 melanoma patients treated with immune checkpoint inhibitors and BRAF/MEK inhibitors, showed that tumor EXOs carrying PD-L1 had immunosuppressive properties since they were as efficient as the melanoma cell at inhibiting T-cell activation. This study provided a rationale for monitoring the EXO-PD-L1 level as a potential predictor of treatment response in melanoma patients [102].

Functional nucleic acids contained in EXO (miRs, long non-coding RNAs, circulating RNAs, mRNAs, and DNA) are chemically stable and resistant to degradation by RNAases. Numerous previous studies have shown that exosomal miRs could be potential diagnostic and prognostic biomarkers for melanoma [103]. In particular, González and his group, in one of the first publications that identified circulating exosomal miRs in patients with melanoma, reported that the EXOs from patients with melanoma contain a level of miR-125b that is significantly lower compared to that contained in EXOs from disease-free melanoma patients or healthy controls [104]. Consistently, miR-125b appears to function as a tumor suppressor in melanoma, by direct regulation of c-Jun and MLK3 protein expressions [105,106] and inducing cell senescence [107]. Recently, Huber et al. have described seven miRs, including miR-125b-5p, that were released in extracellular vesicles and associated with myeloid-derived suppressor cells (MDSCs) and resistance to treatment with immune checkpoint inhibitors in melanoma patients [108]. Moreover, it was reported that the regulation of the miR-125b-5p/EIF5A2 axis in melanoma, through the long intergenic non-protein coding RNA 520 (LICN00520), promotes the proliferation, invasion and migration of melanoma cells. The authors suggest that long non-coding RNA (lncRNA) expression level may be possibly evaluated in blood circulation as a therapeutic biomarker [109].

Another report by Tengda et al. reported that the analysis of a panel of five exosomal miRs, in particular miR-532-5p and miR-106b, was able to distinguish patients with or without metastases, patients with stage I–II from patients with III–IV of disease and patients who received pembrolizumab treatment from those not treated [110].

Most studies have shown the potential role of tumor-derived EVs, mainly EXOs, as biomarkers to predict cutaneous malignant melanoma (CMM) outcome and resistance to MAPKis [110,111,112]. A recent work published by Svedman has led to the identification of the EV-derived miRs, let-7g-5p and miR-497-5p, that can be used as putative good predictive biomarkers after MAPKi treatment in metastatic CMM patients [113]. As reported by Lunavat et al., the expression of miR-211–5p is up-regulated upon vemurafenib and dabrafenib treatment in EXOs and melanoma cells. Interestingly, the authors suggested that stable expression of miR-211–5p reduces sensitivity to BRAF inhibition by regulating cellular proliferation. The authors indicate this miR as a promising prognostic biomarker in melanoma [112].

MiR-21, one of the most frequently up-regulated miRs in solid tumors, controls important tumor suppressor genes as well as genes involved in carcinogenesis, such as phosphatase and tensin homolog (PTEN), programmed cell death protein 4 (PDCD4), phosphoinositide 3-kinase (PI3K), Sprouty and reversion inducing cysteine rich protein with Kazal motifs (RECK) [114,115,116,117,118]. As a result, miR-21 has been proposed as a plausible diagnostic and prognostic biomarker, as well as a therapeutic target for several types of cancer [119]. Moreover, the use of exosomal miR-21 as a potential biomarker has been proposed. Specifically, a systematic review provided evidence that is consistent with this hypothesis although the limited number of each cancer type analyzed should be considered. In addition, the author suggested that a combination of miR panels or cancer antigens may be a good strategy to have better diagnostic results or prognostic predictions in most circumstances [120].

MiR-21 plays a pivotal role in melanomagenesis and is released in EXOs produced from melanoma cells [121]. In an interesting work, Pfeffer et al. performed a miR profiling on RNA prepared from plasma-derived EXOs from specific patient cohorts [122]. As stated in the cited review [120], they found that a panel of five miRs (miR-17, miR-19a, miR-21, miR-126, and miR-149) was expressed at higher levels in plasma-derived EXOs from patients with metastatic melanoma. In addition, these miRs were found to be over-expressed in patients with metastatic sporadic melanoma compared to familial melanoma patients or healthy controls. This panel is proposed as a predictive biomarker to monitor remission as well as relapse following therapeutic intervention [122]. Of interest, miR-21 and miR-146a up-regulations were found in vitreal EXOs of patients with uveal melanoma (UM) as a consequence of dysregulation arising from tumor cells. Data shown in that paper also suggest the possibility of detecting these miRs in VH (vitreous humor) and that serum of UM patients could be considered a potential circulating marker [123]. In addition, the plasma level of miR-21 was positively correlated with tumor burden in metastatic melanoma patients [124] and its relative expression was positively correlated with the tumor nodes and metastasis (TNM) stage of melanoma [125]. All these studies confirmed that the expression of miR-21 in melanoma patients can be used as a potential serum biomarker for melanoma metastasis diagnosis and disease assessment.

Several proteins involved in melanoma progression and metastasis have been identified in EXOs and could be considered as possible prognostic biomarkers [126].

A large increase in total exosomal protein derived from malignant melanoma occurs during disease progression. Specifically, each melanoma cell line appears to exhibit an individual type of heat shock protein (HSP)70 expression, likely reflecting a selection during tumor progression and therapy [127]. Very interesting results were obtained from Peinado and coworkers who identified a melanoma-specific exosomal signature in blood from patients with stage IV melanoma that included expression of tyrosinase-related protein-2 (TYRP-2), very late antigen-4 (VLA-4), HSP70, HSP90 and mesenchymal epithelial transition (MET) oncoprotein [128]. More recently, Alegre et al. evaluated the presence of the melanoma biomarkers melanoma inhibitory activity (MIA) protein, S100B and tyrosinase-related protein 2 (TYRP2) in EXOs derived from serum obtained from stage IV melanoma patients, melanoma-free patients and healthy controls. They observed that MIA and S100B can be detected in EXOs and their quantification presents diagnostic and prognostic utility in melanoma patients [129].

Recently, lncRNAs have emerged as an important regulatory factor in tumor growth and transformation [130]. In particular, Cantile et al. have analyzed the fundamental role played by HOTAIR in the malignant transformation and progression of melanoma cells. Moreover, they have identified this lncRNA in the blood and have suggested its potential role as circulating marker [131]. Numerous reports have shown that exosomal lncHOTAIR could be a sensitive liquid biomarker of different types of cancers, such as glioblastoma multiform, breast cancer and non-small-cell lung cancer (NSCLC) [132,133,134]. However, the potential use of EXOs for HOTAIR detection in melanoma cells has not yet been explored.

In a very recent and interesting study, the authors describe a comprehensive proteomic analysis of 426 human samples from tissue explants (TEs), plasma and other body fluids. They reported that the “tiny packages of materials released by tumors”, called EVPs (extracellular vesicles and non-vesicular particles <50 nm), may serve as biomarkers for detecting a number of different types of cancers (including melanoma) in its early stages. In particular, the analysis of some specific proteins (e.g., versican, tenascin C, and thrombospondin 2 can distinguish tumors from normal tissues with 90% sensitivity and 94% specificity. Moreover, they have defined a panel of tumor-type-specific EVP proteins in TEs and plasma, which can classify tumors of unknown primary origin [135].

## 3. Conclusions

Advanced melanoma is a rapid metastasizing and drug-resistant cancer [1,2]. Despite the availability of new therapies discovered in recent years, metastatic melanoma remains a highly lethal disease. Recent advances in nanotechnologies opened up new perspectives for treating melanoma, potentially effective in overcoming the limitations of conventional approaches: toxicity to normal tissues, development of drug resistance and rapid blood clearance. Nanotechnologies are based on the use of different polymers whose specific characteristics are adapted to different aspects of therapies. Both the Food and Drug Administration (FDA) and European Medicines Agency (EMA) recently approved several types of nanoparticles as a system for conveying nucleic acids, drugs and other molecules for cancer and other disease treatment, including melanoma [17,18,28]. More optimized nanocarriers for the delivery of drugs in a specific way that is able to make the melanoma treatment more efficient is represented by targeted nanoparticles. Melanoma-specific antibodies are conjugated to nanoparticles, thus overcoming the physiological barriers and conditions to reach the target and to recognize and actively bind to target cells through specific antibody–receptor interactions [66].

The discovery of EXOs as natural carriers of RNA, DNA, proteins and lipids have highlighted the potential use of these vesicles as circulating molecular biomarkers for both diagnostic and therapeutic purposes. Due to their easy handling and their ability to transfer molecules between cells, EXOs may be a very performant delivery system [86]. Moreover, naturally secreted EXOs loaded with therapeutic molecules may optimize efficacy of treatment while also reducing off-target delivery. Specifically, molecules may be included in EXOs that are difficult to deliver intracellularly without the use of a carrier (ncRNA, recombinant proteins, small-molecule drugs). Particularly in melanoma, EXOs are also studied as a therapeutic cell-free vaccine [95]. The purification of pure populations of EXOs from EXO-secreting cell lines, unlike those released from autologous primary cells, have immunogenic and oncogenic potential. However, the modification of the complex structure of naturally secreted EXOs may limit their functional activity and their pharmaceutical acceptability. As a potential alternative, some authors have proposed the development of EXO-mimetics manufactured using selected components of EXOs that can be incorporated into synthetic nanoparticles to enhance their stability, immunogenicity, targeting, and uptake [136]. However, the main and useful components of exosomes to therapeutic delivery have not been identified yet.

Although the high number of publications in this field suggests the high diagnostic and prognostic potential of EVs in melanoma, our knowledge remains limited. Several factors hinder the real use of EVs or EXOs as biomarkers. For example, the differences in the methods used by researchers to isolate EXOs from patients’ blood introduce heterogeneity in the composition of the different preparations, thus precluding the identification of the active components. Furthermore, different populations of vesicles are found in the plasma of patients and the isolation of those released by melanoma tumor cells would be of great scientific importance. In this regard, a recent publication by Ferrone et al. suggests the use of an antibody directed against CSPG4 antigen to separate exosomes released by melanoma from those secreted by normal cells [137].

In conclusion, natural or synthetic nanoparticles are promising tools to design new therapeutic, diagnostic and prognostic approaches to cancer and, in particular, to melanoma. Studies are needed to fill the gap between the large amount of evidence available in the preclinical setting and the limited evidence obtained in the real clinical situation.

## Figures and Tables

**Figure 1 cancers-12-02893-f001:**
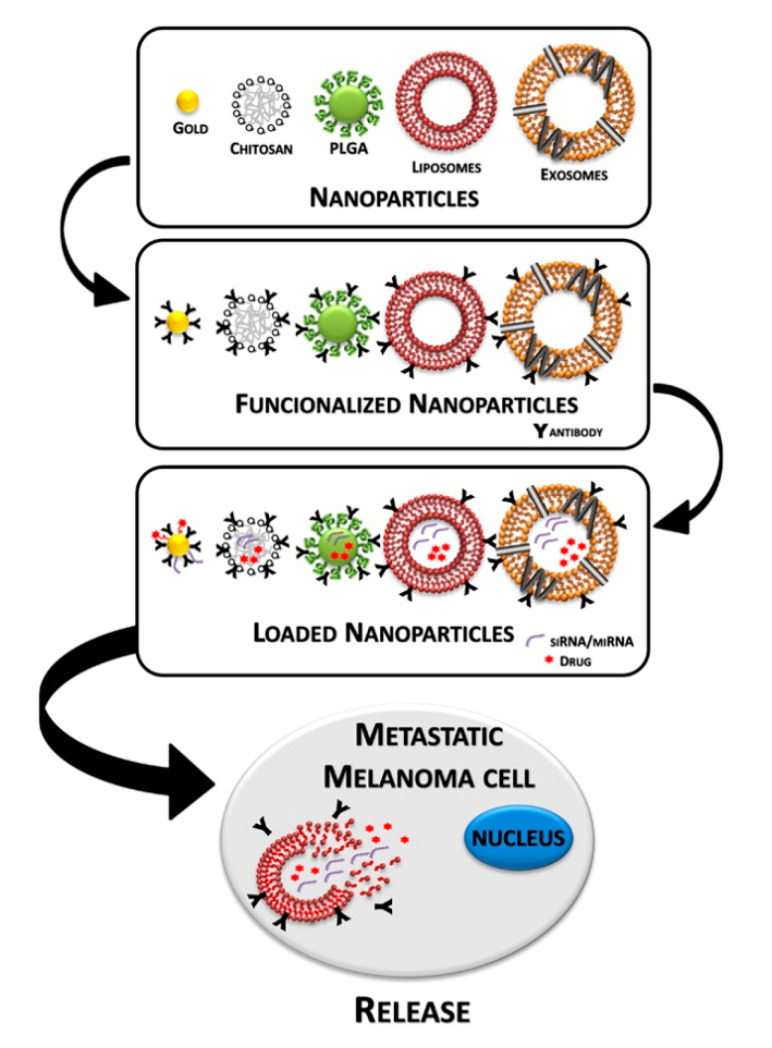
Step-by-step representation of a nanoparticles-based delivery system for melanoma therapy.

**Table 1 cancers-12-02893-t001:** Several types of nanoparticles used in melanoma treatment.

Nanoparticle	Results	Cell Line Used	Reference
Albumin hybrid nanoparticles	Better tumor-targeting capacity and significantly increased drug accumulation in tumor	B16F10 melanoma-bearing mice	[58]
Micelles	High loading efficiency of drug	B16F10	[59]
Dendrimers (PAMAM DAB, PEA) grafted with PEG, acetyl groups, carbohydrates	Increased the bioavailability and efficiency of transported compounds	B16F10	[60]
Copper nanoparticles	Induced cell death by inducing oxidative stress	B16F10 melanoma-bearing mice	[61]
Iron oxide nanoparticles	All nanoparticles induced selective toxicity and caspase 3 activation through mitochondria pathway; Caused generation of ROS, mitochondrial membrane potential decline, mitochondria swelling and cytochrome c release	F10	[62]
Carbon nanotube			
Nanoemulsions (Coffee oil- algae oil-based)	Effective inhibition of melanoma cell growth; Cell cycle arrested at G2/M phase	B16F10	[63]
Nanoemulsion of 5-FU	Much more efficacious than free 5-FU when used for topical delivery	SK-MEL 5	[64]
Multi-peptide and toll-like receptor 4 agonist co-delivery system based on lipid coated Zinc-phosphate hybrid nanoparticles	Exhibited anti-tumor immunity evident by secretion of cytokines *in vitro* and increased CD8+ T-cell response from IFN-γ ELISPOT analysis ex vivo; Improved anti-tumor effects evidenced from prophylactic, therapeutic and metastatic melanoma tumor models compared with free antigens and single peptide-loaded nano-vaccines	B16F10	[65]

Abbreviations: PAMAM, poly-amidoamine; DAB, diaminobenzidine; PEA, palmitoylethanolamide; PEG, polyethylene glycol; ROS, reactive oxygen species; 5-FU, 5-Fluorouracil; IFN-γ, interferon-gamma.

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
