# Peer review of "Advances in Natural or Synthetic Nanoparticles for Metastatic Melanoma Therapy and Diagnosis"

_cancers, 2020, doi:10.3390/cancers12102893_

Round 1

Reviewer 1 Report

1. Authors cited more than 100 scientific papers in this review, ranging from 1999-2020. The "recent" papers (my definition is in three years, i.e. 2018 to 2020) are not the majority, however. Authors may change the title "Recent advances" to reflect the reality better.

2. LINW 178, "TAT protein" should be TAT peptide to reflect "CPP" better.

3. Remove LINE 448-452 "LK    http://................." Not sure why this is here.

Author Response

Response to Reviewer 1 Comments

Point 1: Authors cited more than 100 scientific papers in this review, ranging from 1999-2020. The "recent" papers (my definition is in three years, i.e. 2018 to 2020) are not the majority, however. Authors may change the title "Recent advances" to reflect the reality better.

Response 1: According to the reviewer’s suggestion, the title of our manuscript has been revised and changed in “Advances in natural or synthetic nanoparticles for metastatic melanoma therapy and diagnosis”.

Point 2: LINW 178, "TAT protein" should be TAT peptide to reflect "CPP" better

Response 2: Thanks for this indication. We edited this mistake.

Point 3: Remove LINE 448-452 "LK    http://................." Not sure why this is here

Response 3: Thanks for this indication. We edited this mistake.

Reviewer 2 Report

only minor correction to English style/spelling suggested.

in reviewer's opinion the article is interesting for researchers interested in the field of nanoparticles for drug delivery

Author Response

Response to Reviewer 2 Comments

Point 1: Only minor correction to English style/spelling suggested. in reviewer's opinion the article is interesting for researchers interested in the field of nanoparticles for drug delivery

Response 1: According to the reviewer’s suggestion, the manuscript has been revised by an English speaking person.

Reviewer 3 Report

The manuscript by Arasi et al summarizes the knowledge on both artificial and natural nanoparticles in melanoma treatment. In the first part, the authors describe the different types of artificial nanoparticles, which can be used for melanoma therapy. They give a general overview, but their description is somewhat superficial and the part needs more detailed descriptions of the individual findings.  In the second part, natural nanoparticles are described, but the authors limit themselves to particles derived from the melanoma itself, and to diagnostic use of those. Ideas on the therapeutic use of natural nanoparticles, derived from other cells as the melanoma, are not mentioned. The authors should point that out, and could refer to a recent review in cancers by Cilla et al. (PMID: 32630815). Nevertheless, the review is interesting, and solid.

Minor: The formatting and spelling needs some workover (e.g the abbreviation for interferon is IFN, not INF), abbreviations should be properly defined upon first mentioning. The references are not properly formatted.

Author Response

Response to Reviewer 3 Comments

Point 1: The manuscript by Arasi et al summarizes the knowledge on both artificial and natural nanoparticles in melanoma treatment. In the first part, the authors describe the different types of artificial nanoparticles, which can be used for melanoma therapy. They give a general overview, but their description is somewhat superficial and the part needs more detailed descriptions of the individual findings.  In the second part, natural nanoparticles are described, but the authors limit themselves to particles derived from the melanoma itself, and to diagnostic use of those. Ideas on the therapeutic use of natural nanoparticles, derived from other cells as the melanoma, are not mentioned. The authors should point that out, and could refer to a recent review in cancers by Cilla et al. (PMID: 32630815). Nevertheless, the review is interesting, and solid.

Response 1: The manuscript has been modified according to the reviewer’s comments and suggestions. In the first part, a more detailed descriptions of the individual findings on therapeutic use of artificial nanoparticles have been included. In the second part, additional data relative to therapeutic use of natural nanoparticles derived from not only melanoma cells have been included in the review. Moreover, ideas on the therapeutic use of some particular cells used as natural nanoparticles are now mentioned. Some additional references have been included in the revised paper.

Point 2 (Minor): The formatting and spelling needs some workover (e.g the abbreviation for interferon is IFN, not INF), abbreviations should be properly defined upon first mentioning. The references are not properly formatted.

Response 2: Thanks for this indication. We edited this mistake. The manuscript has been modified according to the reviewer’s comments.